# COVID-19 Pandemic and International Students’ Mental Health in China: Age, Gender, Chronic Health Condition and Having Infected Relative as Risk Factors

**DOI:** 10.3390/ijerph19137916

**Published:** 2022-06-28

**Authors:** Collins Opoku Antwi, Michelle Allyshia Belle, Seth Yeboah Ntim, Yuanchun Wu, Emmanuel Affum-Osei, Michael Osei Aboagye, Jun Ren

**Affiliations:** 1Department of Psychology, Zhejiang Normal University, Jinhua 321000, China; cantwi28@zjnu.edu.cn (C.O.A.); michellebelle15@zjnu.edu.cn (M.A.B.); a1429346851@zjnu.edu.cn (Y.W.); 2International Institute for Child Studies, Zhejiang Normal University, Hangzhou 310012, China; nana2ntim@zjnu.edu.cn; 3KNUST School of Business, Kwame Nkrumah University of Science & Technology, Kumasi AK-039, Ghana; eaffum-osei@knust.edu.gh; 4Department of Interdisciplinary Studies (DIS), Akenten Appiah-Minka University of Skills Training and Entrepreneurial Development, Kumasi AK-039, Ghana; moaboagye@aamusted.edu.gh

**Keywords:** COVID-19 pandemic, international students, stress, anxiety, depression

## Abstract

International students in China were among the first group of individuals to be affected by the COVID-19 pandemic. However, the pandemic’s impact on their mental health is underexplored. This study—utilizing web-based survey data (N = 381), presents preliminary reports using ANOVA and MIMIC analytic approaches. Following the clinical demarcation of the 21-item version of the Depression Anxiety Stress Scales (DASS-21), we found 24.6%, 38.3%, and 43.6% of the students to suffer mild to extreme stress, anxiety, and depression, respectively. Female students reported significantly higher levels of stress and depression than males. Older students’ reports of stress were more substantial than younger students. Students who reported having a relative infected with the virus (vs. those without) experienced significantly higher anxiety and stress. Those who reported having pre-existing chronic health condition(s) (vs. those without) also reported significantly higher stress, anxiety, and depression levels. Moreover, students with an exercise routine (vs. those without) experienced significantly lower levels of stress, anxiety, and depression. Last, our MIMIC model results indicate that foreign students’ age, gender, chronic health status, and having a relative infected with the virus constitute significant risk factors explaining variations in foreign students’ experience of psychological distress. Implications for international students’ management have been thoroughly discussed.

## 1. Introduction

The COVID-19 pandemic in early 2020 threw the world into chaos. It shook the very foundations of our lives: economic collapse, nationwide quarantine or curfew implementation, and intrusive health checks. The global reach [1] of this invisible “demon” meant that every individual, community, and nation had to grapple with its dramatic ramifications. Adaptation to the sudden and rapid change of events was critical to our survival. One key component that marked adaptation measures at the individual, community, and national, or global level was the need for social distancing [2,3]. Like other service industries such as hospitality, travel, and tourism (HTT) [4,5,6,7,8], universities depend on social interaction, for the most part, to contribute to society. However, the high transmissibility of the disease and the attendant urgency to defeat it through breaking the chain (i.e., social distancing) [9,10] completely transformed the character of academic engagements, especially those between students and their faculty [11,12]. Virtual teaching and learning and research have replaced the in-person face-to-face organization of universities’ life [13,14]. 

Additionally, students who lived on campus were urged to eschew all forms of socialization. These developments have critical implications on students’ wellbeing and overall quality of life, particularly international students who are away from their families and often other familiar social groups [15]. In 2022, though the chaotic responses have died down a bit, the disease unfortunately still lives amongst us, mutating into more virulent variants in some countries [16,17]. In addition, in most parts of the world, the new normal dictated by the need for social distancing still prevails.

Available evidence on COVID-19 demonstrates that the pandemic has affected the mental health of patients and survivors and health workers, and the general public [18,19]. Preliminary survey studies targeting people with or without quarantine management status demonstrated the pandemic’s effect on mental health [20,21,22]. Importantly, Vindegaard and Benros [18], through a systemic review of 43 studies, synthesized current evidence on mental health. The authors discovered that 20 studies found deteriorating mental health among health workers, including increased depression/depressive symptoms, anxiety, and poor sleep quality. Nineteen of these studies reveal a similar pattern of mental health degradation among the general public. Nonetheless, research into mental health management in times like this is lacking. The authors, therefore, recommend more studies into mental health to inform the design of preventative and treatment measures and the overall mental health care planning in this and future pandemics.

One important group that academics have been silent on is international students. For instance, international students in China have experienced first-hand the drastic measures that had to be taken to curb the spread of the disease. For many, this included severe restrictions on movement, online classes rather than the in-classroom setting, decline or halt in socialization, and frequent health-related tests and checks right from the start of the pandemic. International students might have been faced with immense socio-emotional pressure as a result. Added to the possible anxiety brought about by fear of contracting the disease [23], they are likely to also worry over the welfare of loved ones across the oceans [24]. Generally, research evidence demonstrates the mental health impacts of the pandemic on students [25,26,27,28,29]. Some examples include survey reports of psychological distress among Indian undergraduates and postgraduate students [30], medical students in Pakistan [31], and Chinese university students [32,33,34]. Specifically, Jiang [32] reports a significant increase in scores of somatizations, obsessive-compulsive disorder, interpersonal sensitivity, anxiety, phobic anxiety, and paranoid ideation, among other mental health indices for Chinese students. It is noteworthy that these students possessed limited COVID-19 knowledge and perceived the risk of infection very highly. All these are factors that likely lead to a coping style that is debilitating to mental health. Now, suppose Chinese university students with close family ties [35] are in the throes of COVID-19. In that case, we can safely assume that international students—who may have language challenges, limited mobility, and cultural understanding, may not be spared. 

However, scant research exists on international students whose lives, from all indications, may be more susceptible to bouts of stress, anxiety, and depression. Indeed, a systematic review of research on the mental health of international students in China from 2014 to 2021 by Wu [36] found that only two studies examined international students since the pandemic, of which one explored barriers to health science international students’ online education [37] and the other on psychological determinants of depression [38]. These studies are important, but they do not reveal the nature of the incidence of psychological distress across subpopulations of international students. Moreover, Kulyar’s [24] letter to the editor suggests that international students’ concern about infection and safety of their families may impact their mental health. Wang’s preprint [39] with a sample from Central South University’s international students indicates the prevalence of depression and anxiety. These worrying mental health indications are not surprising. Pre-COVID-19 research evidence illustrates the various psychological distresses of adaptation challenges faced by international students in China [40,41,42,43]. Wu [36] therefore concluded that the psychological situation of international students in China is unsatisfying but limited research exists in this area. Manzoor’s [23] recent study on how the fear of COVID-19 induces generalized anxiety disorder among foreign students in China is a welcome contribution. 

That notwithstanding, it is important to estimate the extent of mental health incidence among international students. Accordingly, the first objective of this study is to examine the international students’ experience of psychological distress (i.e., stress, anxiety, depression) during the COVID-19 pandemic. Moreover, it is essential to acknowledge that international students are not monolithic. They are some of the most diverse social groups; they have different funding sources, academic majors, and accommodation preferences among other things. To get a comprehensive picture of mental health incidence among international students, our second objective is to analyze international students’ experience of psychological distress across subpopulations using their socio-demographic profiles. Last, we build a multiple indicator-multiple cause (MIMIC) model to examine the predictive capacity of relevant international students’ profiles (i.e., age, sex, chronic condition, and relative infected with COVID-19). In sum, this study examines: (1) the extent of international students’ experience of psychological distress; (2) the levels of this experience across subpopulations; and (3) the risk factors.

## 2. Materials and Methods

### 2.1. Participants and Procedure 

This study adopts a cross-sectional design, using a questionnaire survey, in assessing international students’ mental health in a time of infectious disease outbreak—the COVID-19 pandemic. The survey was conducted when international students in most schools were still unable to move in and out of campus and were confined to their hostels or dormitories. And the few who could step out of the walls of their universities did so within a strict 1 to 3 h timeframe. Such mobility limitations are currently implemented in universities in certain cities such as Shanghai. The study’s goal and the need for social distancing demanded that an online survey questionnaire design be used for data collection. So, a convenience sampling approach was adopted. The questionnaire items for the constructs (i.e., the DASS-21 variables) and students’ profiles explored in this study were organized on an online platform called Kwiksurvey.com (this platform is used for individual and corporate data collection purposes. For more details, please consult https://kwiksurveys.com/). A hyperlink and a quick response code (i.e., QR code) of the online survey were generated from the platform. To reach a diverse international student population, the authors joined many international student groups on WeChat (a Chinese version of Facebook). Then, we introduced the study along with the hyperlink and the QR code, the study’s purpose, and the supervising institution. To encourage participation, researchers presented a gift packet of money called HongBao to the group members. Data collection took place in July of 2020. The final usable sample for this study came to 381. Eighteen (18) cases were deleted from the data due to incomplete responses, constituting 4.5% of the total data collected.

### 2.2. Measures 

*Mental health variables*: The three variables measuring international students’ mental health—depression, anxiety, and stress, were captured using the 21-item version of the Depression Anxiety Stress Scales (DASS-21) [44]. These variables were each measured with 7 items on the DASS-21. Example items include: “I was aware of the dryness of my mouth” for anxiety [α = 0.82], “I felt down-hearted and blue” for depression [α = 0.90], and “I found it hard to wind down” for stress [α = 0.84]. The students were instructed to respond to the items in accordance with how best each statement described their feeling in the week prior. Students’ responses were recorded on a 4-point Likert-style scale (0 = *did not apply to me at all* to 3 = *apply to me most of the time*). 

*Students’ demographics and lifestyle variables*: By the time of the data collection, COVID-19 had literally enveloped the globe. Therefore, relevant demographic and lifestyle profiles were captured to see where students’ experience of psychological distress differs across these salient indicators. Close- and open-ended questions were used to obtain students’ profiles, consisting of gender (1 = *male* and 2 = *female*), age (respondents were asked how old they were at their last birthday and then transformed into categorical variable: 1 = ≤24; 2 = 25–29; 3 = 30–34; 4 = 35 and above), study level (1 = *Chinese language study*; 2 = *college/university*; 3 = *master*; 4 = *PhD*), residence type (1 = *on-campus accommodation* and 2 = *off-campus accommodation*), completion year (1 = 2020; 2 = 2021; 3 = 2022; 4 = 2023; 5 = 2024; 6 = 2025), sponsorship (*self-finance*; 2 = *full scholarship*; 3 = *other*), economic situation (1 = *very good*; 2 = *fairly good*; 3 = *fairly bad*; 4 = *very bad*), relative on the frontline (1 = *yes* and 2 = *no*), relative infected with COVID-19 (1 = *yes* and 2 = *no*), chronic health condition (1 = *yes* and 2 = *no*), exercise routine (1 = *yes* and 2 = *no*), exercise time per week (measured in minutes and then transformed into categories of 1 = ≤2 h; 2 = 2.1–4 h; 3 = >4 h), years of stay (measured in months and then transformed into categories: 1 = ≤3 years; 2 = 3.1–6 years; 3 = >6 years). Please consult Table 1 for details on the descriptive analysis of students’ profiles. 

### 2.3. Analytic Approach

This study performed descriptive analysis to obtain frequencies and stacked percentage bar charts for students’ sociodemographic variables and DASS subscales (i.e., anxiety, depression, and stress). Further, a one-way analysis of variance (ANOVA) was used to compare students’ characteristics and psychological states (i.e., anxiety, depression, and stress). Last, we built a multiple indicator-multiple cause (MIMIC) model [45,46] to explore the effect of students’ characteristics (i.e., gender, age, relative with COVID-19, and chronic health conditions) on anxiety, depression, and stress. SPSS version 25 and MS Excel were used to perform all the statistical analyses.

## 3. Results

### 3.1. International Students’ Sociodemographic Distribution

The majority of the students surveyed were studying towards a bachelor’s degree or higher (91.8%). Females (50.7%) were only slightly represented more than males. The age distribution of the students showed that they were predominantly below or at 29 years old (73.3%). The greater mass of the students lived in an on-campus accommodation (68.0%), were scheduled to graduate at least by 2021 (64.6%), followed an exercise routine (61.4%), and exercised at least 2 h per week (54.6%). On the other hand, fewer students recorded having a chronic health condition (6.0%), a relative as a frontline worker (25.7%), or a relative with COVID-19 (10.2%). Given that the majority of the students had full scholarship (63.3%), it is unsurprising that the economic situation of the more significant number of the students was at least fairly good (64.5%) (see Table 1 for more details). 

### 3.2. International Students’ Experience of Mental Health during COVID-19

This study took a snapshot of mental health incidence among international students in China. Generally, the findings presented in Figure 1**.** illustrate a concern for the effects of the COVID-19 pandemic on international students’ wellbeing. The clinical demarcation of mental health experience follows the scoring chart of the DASS-21 scale as detailed in [44]. Overall, over 38 percent of the students surveyed reported experiencing mild to severe anxiety. Moderately anxious feelings were the highest (12.9%), with severe anxiety being the least (6.8%). The percentage of stressed-out students was generally low (24.6%), with only 2.6% of them experiencing extreme stress levels. In terms of depression, the distribution mirrors anxiety, with more (43.6%) of the respondents experiencing varying levels of depression. However, unlike anxiety, the students’ experience of mild (15.0%) and moderate (15.0%) depression was the same (see Figure 1 below). 

### 3.3. ANOVA Results for the Different Groups of Students’ Demographic and Lifestyle Variables

A one-way ANOVA analysis results revealed some sociodemographic groups differ significantly in their experience of stress, anxiety, and depression. Levene’s homogeneity test of variance was significant for gender groups at *p* = 0.002 for depression but non-significant at *p* = 0.125 for stress. Female international students reported experiencing significantly higher levels of depression [*F* = 8.22, *p* < 0.05] and stress [*F* = 7.31, *p* < 0.05] compared to their male counterparts using the Brown–Forsythe test. In respect of age, Levene’s test of homogeneity of variance was observed to have significance at *p* = 0.022 for stress and *p* = 0.045 for depression. However, only stress levels differed significantly across the age groups [*F* = 3.03, *p* < 0.05]. Levene’s homogeneity test of variance was examined for students’ exercise routine and found significance at *p* = 0.040 and *p* = 0.004 for anxiety and depression, respectively, but not stress *p* = 0.098. Hence, students’ exercise routine status shows itself as a significant differentiating variable in determining the levels of student experience of anxiety [*F* = 9.67, *p* < 0.05], depression [*F* = 9.86, *p* < 0.05], and stress [*F* = 4.02, *p* < 0.05] using the Brown–Forsythe robust test. Further, Levene’s homogeneity test of variance was examined for groups with a relative with COVID-19 and was significant at *p* = 0.006 and *p* = 0.005 for anxiety and stress, respectively. A significant difference was found between groups with and without a relative with COVID-19 regarding anxiety [*F* = 7.33, *p* < 0.05] and stress [*F* = 3.94, *p* < 0.05]. Lastly, significance was found for Levene’s test of homogeneity of variance for groups with and without chronic health conditions at *p* = 0.002, *p* = 0.000, and *p* = 0.016 for anxiety, depression, and stress, respectively. A significant difference was recorded between a student with and without chronic health conditions concerning anxiety [*F* = 6.97, *p* < 0.05], depression [*F* = 11.21, *p* < 0.05], and stress [*F* = 12.71, *p* < 0.05] using the Brown–Forsythe robust test. No statistical significance was found between groups of other sociodemographic variables surveyed in Table 2 with respect to the DASS subscales.

### 3.4. MIMIC Results

MIMIC model was employed to assess the impact of foreign students’ characteristics (i.e., gender, age, relative with COVID-19 and chronic health conditions) on their depression, anxiety, and stress. Prior to the model development, gender (female = 0 and male = 1), relative with COVID-19 (no = 0 and yes = 1), and chronic health conditions (no = 0 and yes = 1) were dummy coded. The model provided a good fit to the data (X2 = 464.70, df = 214, X2/df = 2.17, NFI = 0.89, CFI/TLI = 0.94/0.93, RMSEA = 0.06, SRMR = 0.04). The results revealed that gender had a significant negative effect on anxiety (b = −0.11, SE = 0.03, *p* < 0.05), depression (b = −0.17, SE = 0.08, *p* < 0.01), and stress (b = −0.16, SE = 0.05, *p* < 0.01). Age had a significant positive effect on depression (b = 0.11, SE = 0.01, *p* < 0.05) and stress (b = 0.16, SE = 0.05, *p* < 0.01). Moreover, students with relatives infected with COVID-19 significantly predicted anxiety (b = 0.18, SE = 0.06, *p* < 0.01), depression (b = 0.13, SE = 0.12, *p* < 0.05), and stress (b = 0.14, SE = 0.10, *p* < 0.01). Last, chronic health condition had a significant positive effect on anxiety (b = 0.20, SE = 0.08, *p* < 0.01), depression (b = 0.21, SE = 0.16, *p* < 0.001), and stress (b = 0.26, SE = 0.13, *p* < 0.001) (see Figure 2).

## 4. Discussion

Sadly, the COVID-19 pandemic has engulfed our globe with unprecedented fear, panic and deaths. It has been over two years now since the disease was declared a global health crisis. Unfortunately, catastrophes of this magnitude are marked to debilitate peoples’ physio-psychological health. In this pandemic, several studies have reported the mental health impact of the COVID-19 pandemic among general populations, medical practitioners, and students. However, scant research [36] exists on one group of university students—the international students—who may be more prone to psychological distress. In particular, international students in China are some of the world’s population to first experience the brunt of the COVID-19 pandemic. Accordingly, the primary aim of this study is to offer a preliminary account of the mental health impacts of the COVID-19 pandemic on international students in China. To achieve our purpose, we employed Lovibond and Lovibond’s (1995) DASS-21 scale and its accompanying clinical scoring chart to map out international students’ experience of psychological distress. Additionally, a one-way analysis of variance (ANOVA) was performed to establish whether differences in mental health experience across international students’ sociodemographic variables were statistically significant. Lastly, a multiple indicator multiple causes (MIMIC) model was developed to explore relevant students’ sociodemographic (i.e., age, gender, chronic condition, and relative with COVID-19) as risk factors. Thus, to the best of our knowledge, this study forms the first assessment of international students’ mental health experience across their profiles and the predictive influence of relevant profiles on their experience. 

The results demonstrate that international students’ mental health is a concern. It is noteworthy, however, that a cursory browse through the stacked percentage bar chart indicates that the percentages of students reporting either being anxious, depressed, or stressed out tend to plummet with increasing levels of severity, except anxiety, where the highest rate of students reported feeling moderately (rather than mildly) anxious. This distribution mirrors the distribution of college students’ mental health reports on the DASS-21 scale in prior studies during the COVID-19 pandemic [47]. However, compared to similar studies from China using Chinese university student samples [48], international students report a relatively higher experience of psychological distress. It is also noteworthy that these studies used medical students, who, given the odds, are anticipated to be more psychologically distressed. For example, over 38% of international students in this study reported feeling mild to extreme anxiety compared to 24.9% reported among medical students [48]. Further assessment reveals that a significant part (21.3%) of the 24.9% experienced mild anxiety leaving a small part experiencing moderate (2.7%) and severe (0.9%) anxiety. Moreover, Liu et al. (2020) found that 22.1% and 35.5% of medical students from Hubei province experience anxiety and depression, respectively. In the present study, 43.6% reported feeling depressed. Though Cao et al. [48] and Liu et al. [49] utilized the 7-item Generalized Anxiety Disorder Scale (GAD-7), the exactness of the number of items and the identical rating (thus, the 4-point Likert scale) makes it comparable to the anxiety subscale of the DASS-21. 

In terms of biographical data, findings from the ANOVA indicate that female international students are more depressed and stressed out than male international students. This finding is consistent with previous studies using university students’ samples [50,51,52] and even among adolescents [53]. For Hakami et al.’s [51] study, they discovered that junior females who lived alone were particularly at risk of severe to extreme psychological distress. However, not all studies find females to experience significantly different levels of psychological distress in this pandemic. For example, Cao et al. [48] found no difference between males and females in their experience of psychological distress. Further, this study finds that older international students experienced significantly higher stress levels than younger international students. The negative relations between age and stress may be due to the fact that media reports of higher mortality rate among the older population who get infected. In contradistinction, Gaş et al. [50] discovered that younger students were more stressed out than older students. 

Regarding international students’ social, pre-existing health condition, and lifestyle data, the results show that the international student group that reported having a relative infected with COVID-19 was significantly more anxious and stressed out than those who had no relative infected with the virus. The higher psychological distress might stem from the worry about losing someone so close. Perhaps, even a sponsor of one’s education. In addition, international students who reported having a pre-existing chronic health condition reported experiencing significantly higher levels of depression, anxiety, and stress than students without a pre-existing chronic health condition. Scientific communication at the initial stages of the virus made clear that the condition of the infected is more likely to deteriorate into a critical emergency if pre-existing chronic health condition exists. This finding supports the empirical reports of individuals with pre-existing psychological conditions suffering more psychological distress than those without [54,55]. Lastly, students with an exercise routine seem to cope better than those without as they reported significantly lower levels of stress, anxiety, and depression compared to students with no exercise routine. This result is consistent with current research evidence [56,57,58,59,60], demonstrating negative relation between physical activity and psychological distress.

Lastly, the findings of the test of whether international students’ age, gender, chronic condition status, and having a relative infected with COVID-19 predicted variations in our three latent variables (i.e., depression, anxiety, and stress) of mental health strengthen the significance of taking foreign students’ unique profiles into account in managing their psychological reaction to the pandemic. Being a female, older age, with a chronic condition, and having a relative with COVID-19 predisposes foreign students to experience higher levels of psychological distress. Some of these findings are consistent with existing studies concerning determinants of psychological distress in this pandemic in other populations, including psychological distress among Israeli and European populations [61,62]. Specifically, the evidence that female international students and those with pre-existing health issues experience higher levels of psychological distress is in line with the findings from the Israeli population [62]. Again, being female was found to induce higher psychological distress in 27 European countries [61]. However, the younger age Israeli population was less impacted by the pandemic contrary to our findings that older students experience higher psychological distress.

### 4.1. Implications for International Students’ Health Management in Crisis Time

Scholars have suggested the possibility of psychological distress among international students [15]. This study adds to the scant research on international students’ mental health in China [36] by demonstrating the impact of COVID-19 on international students’ mental health, differences in experiencing psychological distress across subpopulations, and the risk factors. The study, therefore, directs the discussion on international students’ mental health from theory-informed speculations to data-driven conclusions. Furthermore, the findings contained in this study possess several practical implications for international students’ mental health management that aligns with the popular proposal of a student-centered approach to international higher education management that nurtures international students’ wellbeing. 

First, female international students feeling more depressed and stressed suggest that any interventions designed to manage the international students’ mental health should recognize that not all interventions may work just because they work for male international students. Additionally, interventions on stress management may not work for students of all age groups. We recommend that university managements check the efficacy of preventative or treatment programs across age groups and male and female international students. Second, international students’ lifestyle (exercise routine in this study) is found to have a significant impact on their mental health in this pandemic. We recommend that universities develop physical activity protocols that ensure safe participation in these activities. Third, international students’ management should know which of their students are at a higher risk of psychological distress based on their medical history. Management should quickly move and identify these individuals and, with their contribution, identify ways to support them. Finally, global catastrophes of this nature imply that some relatives of international students may be affected by the situation. The results that students who had a relative infected with the virus reported higher psychological distress implies that international students’ management offices need to know these students to learn about their unique conditions and devise ways, including mindfulness training [63], to alleviate their anguish. 

### 4.2. Limitations of the Study

Despite the valuable insights offered by this study, some limitations are worth pointing out to guide the interpretation of the study’s findings. First, the study’s cross-sectional design with self-reported data raises the possibility of biased reports. However, self-report data on university students’ mental health has been adjudged an adequate assessment as individuals are in the best position to evaluate their subjective wellbeing [48,64]. Second, given that the design was cross-sectional, a comparative analysis of international students’ mental health across different time points during the pandemic is not possible. Therefore, we recommend that future studies adopt a longitudinal design to gain a much richer insight into mental health among international students. Third, self-selection bias may pose a challenge as the data were collected online without probability sampling. We recommend that future research access the sampling frame in a particular university or combine both off- and on-line data collection approaches to monitor self-selection biases in web-based surveys. Fourth, this study did not take into account the origin countries of these students. We recommend future research consider the respective countries of the students. Finally, international students’ sociodemographics are numerous, and we recommend that future studies explore other relevant profiles such as family relationship and economic status that have not been explored in this study.

## 5. Conclusions

The impact of COVID-19 on international students’ mental health has been underexplored [15,36]. In this study, we report empirical evidence on the mental health impact of the pandemic on international students’ subpopulations. Specifically, the findings show that students who are female, older, with relatives infected with the COVID-19 virus, have pre-existing chronic health condition(s), and are without exercise routine are more susceptible to mental health degradation during the pandemic. We offer a number of practical recommendations to the management of institutions of higher learning with international students’ population to pay attention to the subpopulation of international students that may be more prone to suffer psychological distress from the pandemic as the virus mutates to more or less virulent variants.

## Figures and Tables

**Figure 1 ijerph-19-07916-f001:**
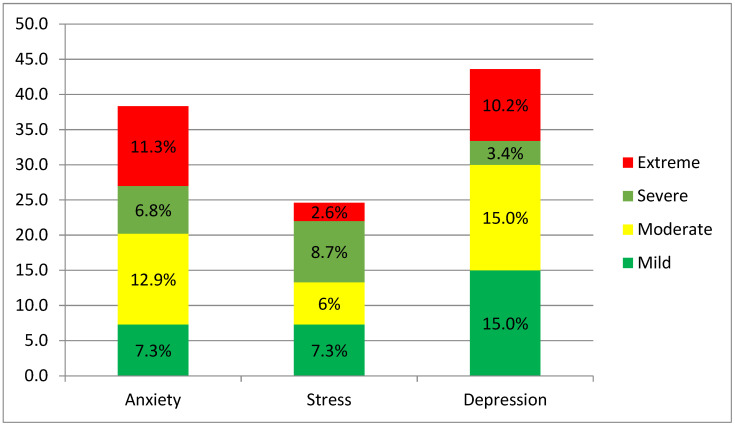
International students’ experience of anxiety, stress, and depression during COVID-19 pandemic in China.

**Figure 2 ijerph-19-07916-f002:**
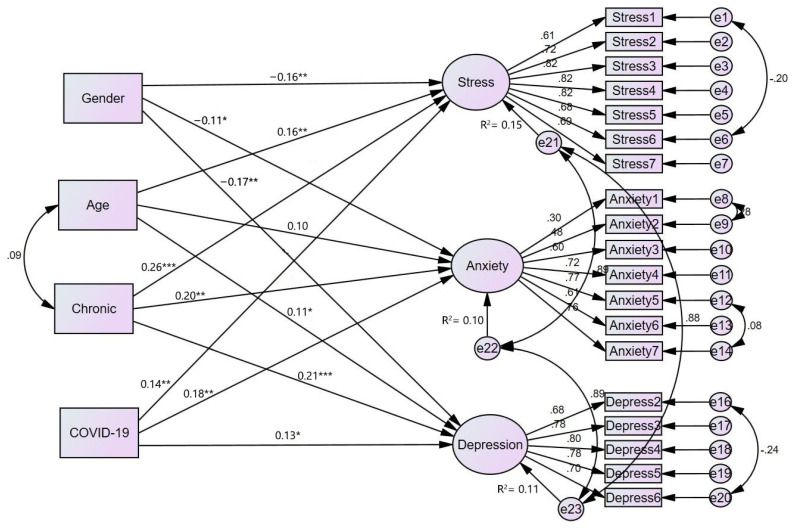
Structural model examining students’ demographics effect on DASS-21. Note: Chronic = chronic health conditions, COVID-19 = relative with COVID-19. * *p* < 0.05, ** *p* < 0.01, *** *p* < 0.001.

**Table 1 ijerph-19-07916-t001:** Descriptive analysis of students’ demographic and lifestyle profiles.

Variable		Freq (%)
Sex		
	Men	188 (49.3%)
	Women	193 (50.7%)
Age		
	≤24	121 (31.8%)
	25–29	158 (41.5%)
	30–34	78 (20.5%)
	35 and above	24 (6.3%)
Sponsorship		
	Self-finance	67 (17.6%)
	Full scholarship	241 (63.3%)
	Others	73 (19.1%
Residence type		
	On-campus accommodation	259 (68.0%)
	Off-campus accommodation	122 (32.0%)
Study level		
	Chinese language	31 (8.1%)
	College/university	160 (42.0%)
	Master	148 (38.8%)
	PhD	42 (11.0%)
Economic situation		
	Good	58 (15.2%)
	Fairly good	188 (49.3%)
	Fairly bad	92 (24.1%)
	Bad	43 (11.3%)
Completion year		
	2020	102 (26.8%)
	2021	144 (37.8%)
	2022	74 (19.4%)
	2023	31 (8.1%)
	2024	14 (3.7%)
	2025	16 (4.2%)
Years of stay		
	≤3 years	234 (61.4%)
	3.1–6 years	119 (31.2%)
	>6 years	28 (7.3%)
Exercise routine		
	Yes	238 (62.5%)
	No	143 (37.5%)
Exercise time/week		
	≤2 h	208 (54.6%)
	2.1–4 h	78 (20.5.8%)
	>4 h	95 (24.9%)
Frontline relative		
	Yes	98 (25.7%)
	No	283 (74.3%)
Relative with COVID-19		
	Yes	39 (10.2%)
	No	342 (89.8%)
Chronic health conditions		
	Yes	23 (6.0%)
	No	358 (94.0%)

**Table 2 ijerph-19-07916-t002:** One-way ANOVA analysis result.

			Anxiety			Depression			Stress	
Variables		Mean ± SD	Levene Statistic Sig.	F	Mean ± SD	Levene Statistic Sig.	F	Mean ± SD	Levene Statistic Sig.	F
Gender										
	Male	6.85 ± 7.78	0.214	2.877	8.80 ± 8.85	0.002 **	8.22 **	9.67 ± 9.25	0.125	7.31 **
	Female	8.27 ± 8.53			11.73 ± 11.02			12.38 ± 10.32		
Age										
	≤24	6.96 ± 7.96	0.358	2.581	9.44 ± 10.67	0.045 *	1.596	9.57 ± 9.78	0.022 *	3.03 *
	25—29	7.94 ± 8.61			10.06 ± 9.44			11.30 ± 9.52		
	30–34	6.56 ± 7.17			10.69 ± 9.34			11.10 ± 9.45		
	35 and above	11.50 ± 8.79			14.67 ± 12.94			16.58 ± 12.42		
Sponsorship										
	Self-finance	6.87 ± 7.71	0.824	0.310	8.17 ± 8.34	0.020 *	1.931	9.70 ± 9.56	0.523	0.801
	Full scholarship	7.68 ± 8.21			10.54 ± 9.95			11.47 ± 9.61		
	Others	7.84 ± 8.19			11.37 ± 11.81			10.88 ± 11.03		
Study level										
	Chinese language	6.19 ± 8.20	0.578	2.256	8.65 ± 9.67	0.084	1.741	9.68 ± 8.84	0.077	0.016
	College/university	6.81 ± 7.67			9.70 ± 9.76			10.36 ± 9.69		
	Master	7.95 ± 8.36			10.39 ± 9.76			11.16 ± 9.63		
	PhD	10.14 ± 9.12			13.33 ± 12.17			14.23 ± 11.78		
Residence type										
	On-campus accommodation	7.52 ± 8.25	0.531	0.028	10.59 ± 10.35	0.231	0.729	11.09 ± 10.18	0.078	0.883
	Off-campus accommodation	7.67 ± 8.09			9.64 ± 9.56			10.95 ± 9.28		
Economic situation										
	Good	6.07 ± 7.57	0.105	1.369	9.00 ± 9.25	0.581	0.897	9.69 ± 8.51	0.334	0.967
	Fairly good	8.09 ± 8.79			10.62 ± 10.55			11.46 ± 10.29		
	Fairly bad	8.07 ± 7.85			11.11 ± 9.96			11.76 ± 10.12		
	Bad	6.28 ± 6.70			8.79 ± 9.50			9.53 ± 9.30		
Completion year										
	2020	8.02 ± 7.76	0.109	0.829	10.90 ± 9.33	0.091	0.707	12.08 ± 8.97	0.002 **	1.060
	2021	7.39 ± 7.99			9.96 ± 9.80			10.56 ± 9.59		
	2022	8.27 ± 9.52			10.70 ± 11.17			11.57 ± 11.94		
	2023	7.81 ± 9.02			11.03 ± 12.26			11.55 ± 11.69		
	2024	4.43 ± 6.19			10.00 ± 11.29			7.57 ± 6.03		
	2025	5.38 ± 5.45			6.13 ± 6.39			8.50 ± 5.34		
Years of stay										
	≤3 years	7.70 ± 8.44	0.110	0.654	10.28 ± 10.24	0.020 *	0.964	10.90 ± 10.12	0.005 **	1.609
	3.1–6 years	7.01 ± 7.45			9.66 ± 9.04			10.52 ± 8.66		
	>6 years	8.86 ± 9.13			12.93 ± 12.89			14.50 ± 12.30		
Exercise routine										
	Yes	6.53 ± 7.56	0.040 *	9.672 **	8.97 ± 9.01	0.004 **	9.857 **	10.26 ± 9.29	0.098	4.021 *
	No	9.30 ± 8.90			12.48 ± 11.40			12.35 ± 10.71		
Exercise time/week										
	≤2 h	8.01 ± 7.91	0.096	1.750	10.73 ± 9.75	0.471	1.143	11.46 ± 9.41	0.236	1.128
	2.1–4 h	8.05 ± 9.18			10.74 ± 11.41			11.54 ± 11.29		
	>4 h	6.21 ± 7.86			8.93 ± 9.69			9.73 ± 9.66		
Frontline relative										
	Yes	8.08 ± 9.23	0.077	0.515	10.14 ± 11.16	0.117	0.025	10.76 ± 10.92	0.066	0.113
	No	7.39 ± 7.81			10.33 ± 9.73			11.14 ± 9.52		
Relative with COVID-19										
	Yes	11.79 ± 10.54	0.006 **	7.325 **	13.74 ± 12.40	0.005 **	3.521	14.72 ± 12.48	0.005 **	3.936 *
	No	7.09 ± 7.75			9.89 ± 9.75			10.63 ± 9.48		
Chronic health conditions										
	Yes	13.57 ± 11.42	0.002 **	6.966 *	19.48 ± 13.81	0.000 ***	11.209 **	19.65 ± 12.09	0.016 *	12.712 **
	No	7.18 ± 7.80			9.69 ± 9.54			10.49 ± 9.48		

SD = standard deviation; * *p* < 0.05, ** *p* < 0.01, *** *p* < 0.001.

## Data Availability

Data will be supplied upon request from the corresponding author.

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
