# Peer review of "COVID-19 Pandemic and International Students’ Mental Health in China: Age, Gender, Chronic Health Condition and Having Infected Relative as Risk Factors"

_ijerph, 2022, doi:10.3390/ijerph19137916_

Round 1
Reviewer 1 Report
This is a very interesting paper focused on mental health outcomes among international students in China.
The authors have improved substantially the paper and followed-up the suggestions I recommended.
However, I suggest to implement minor changes before publishing it in the journal.
In the abstract section, the authors are not reporting the methods and the statistical analyses they performed. I consider it important in order to interpret in a right way the main results.
At the end of the introduction section, the authors are reporting that, to achieve the objectives, they made three key contributions. I recommend to reformulate it in order to draw the main objectives, not the contributions.
In Table 1: It would be prefered to say "men and women" and not "male and female".
Author Response
- On reporting statistical analytic approaches employed in this study in the abstract, we thank the reviewer for this suggestion. We have added this information to the abstract. Please see page 1, lines 18 and 19.
- On reformulation of the end of the introduction section to highlight objectives and not contributions, we thank the reviewer for this suggestion. We have reformulated the end of the introduction section to reflect this suggestion. Please check page 3, lines 116 – 118.
- On writing "men and women" and not "male and female", we thank the reviewer for this suggestion. We have, as advised by the reviewer, replaced "male and female" with "men and women" respectively. Please see Table 1 on page 4, lines 206 – 209.
Reviewer 2 Report
Thank you very much for giving me the opportunity again to review the paper. The paper is noticeably improved. It responds perfectly to my initial assessment as well as to the other reviewer. My congratulations to the authors. I think it should be published.
Author Response
We are grateful to the reviewer for recommending our paper for publication and his/ her warm congratulatory message.
This manuscript is a resubmission of an earlier submission. The following is a list of the peer review reports and author responses from that submission.
Round 1
Reviewer 1 Report
Thank you